

# Estimating the impact of environmental management on strawberry yield using publicly available agricultural data in South Korea

Steven Kim[1,*], Jung Su Jo[2,3,*], Vicky Luk[1,4], Sung Kyeom Kim[2,3] and Dong Sub Kim[5]

[1] Mathematics and Statistics, California State University, Monterey Bay, Seaside, United States
[2] Horticultural Science, Kyungpook National University, Daegu, South Korea
[3] Institute of Agricultural Science and Technology, Kyungpook National University, Daegu, South Korea
[4] Applied Environmental Science, California State University, Monterey Bay, Seaside, United States
[5] Horticulture, Kongju National University, Yesan, South Korea
* These authors contributed equally to this work.

Corresponding authors
Sung Kyeom Kim,
skkim76@knu.ac.kr
Dong Sub Kim,
dongsub@kongju.ac.kr

## ABSTRACT

Advanced information and communication technologies (ICTs) have made data collection more efficient for agricultural studies. Using publicly available database in South Korea, we estimated the relationship between the management of air temperature and relative humidity and the strawberry yield during two harvest seasons. Longitudinal data of multiple greenhouses were merged and processed, and mixed-effects models were applied to account both observed and unobserved factors across the greenhouses. The averages of air temperature and relative humidity inside each greenhouse do not take volatility of the time-varying variables into consideration, so we assessed the management of each greenhouse by the percent of time that air temperature between 15 °C and 20 °C (denoted as $T_\%$) and the percent of time that relative humidity between 0% and 50% (denoted by $H_\%$). The statistical models estimated that the strawberry yield decreases with respect to the number of days since harvest began and the rate of decrease is slower when $T_\%$ and $H_\%$ are higher. This study used large-scale multilocation data to provide the practical suggestion that air temperature and relative humidity should be maintained within the optimal ranges to mitigate the loss of strawberry yield especially at the later phase of a harvest season.

# INTRODUCTION

Advanced ICTs have generated big data in various fields. Big data usually means a large amount of structured or unstructured information that is complex and difficult to process using existing databases and management processes (*Kim & Lee, 2020*; *Manyika et al., 2011*). Agricultural researchers can benefit from big data generated during the cultivation process, and it contains growth variables, environmental conditions, and yields. To utilize

the useful information, it is important to have a centralized data management system, so the South Korean government legislated the Act on Promotion of Provision and Use of Common Data (*Ministry of the Interior & Safety of South Korea, 2013*). The public data are accessible in various formats such as text, figures, and images, and the amount of available data is growing every year. As of December 2021, the public data portal provides access to 938 open institutions, 49,876 data files, 8,381 open application programming interfaces (APIs), and 8,758 standard datasets. Among these sources 2,680 data files, 830 open APIs, and 159 standard datasets are related to agriculture and livestock (*Ministry of Interior & Safety of South Korea, 2021*, *2022*).

Past studies have used environmental data to explain and predict the growth and yield of various crops. The environmental data include information on environmental conditions (air and soil temperatures, relative humidity, radiation, wind direction and speed, *etc.*), cultivation (soil or hydroponics cultivation systems, electrical conductivity (EC) and pH of nutrient solution, *etc.*), and growth and yield (plant height, leaf length and width, numbers of leaves, nodes, flowers, and fruit, and fresh and dry weight of fruit, *etc.*). A number of studies have reported the relationships among environmental conditions, cultivation systems, and growth and yield of crops. For instance, the air temperature and irrigation rate shown to be related to the leaf size (*Wright et al., 2017*), and the daily average air temperature and the temperature difference between day and night are shown to be related to the node length, plant height, and flower stalk elongation (*Myster & Moe, 1995*). Moreover, the relative humidity is found to be related to photosynthesis and fruiting (*Xue, Li & Wen, 2010*), and the light intensity is found to be related to the plant height, fresh weight, and dry weight (*Fan et al., 2013*).

This study focuses on strawberry which is a high-income horticultural crop. Globally, it covers a cultivated area of 384,668 ha, yielding 23,036 kg/ha (*FAOSTAT, 2020*). In South Korea, it covers a cultivated area of 5,683 ha, and 63,646 tons of strawberry were yielded in the 2020 season (*Korean Statistical Information Service (KOSIS), 2020*). Among several cultivars of strawberry, more than 80% of strawberries cultivated in South Korea are the 'Seolhyang' cultivar (*Rural Development Administration of South Korea (RDA), 2022*). Some studies have focused specifically on the effect of environmental factors on strawberry fruit. Air temperature and relative humidity sensors are easy to install and inexpensive to purchase, and they are key environmental factors for strawberry yield which are easily controlled in a greenhouse by ventilation (*Körner & Challa, 2003*). *Kadir, Sidhu & Al-Khatib (2006)* reported that 'Chandler' and 'Sweet Charlie' grown at 20/15°C (day/ night) produced more and larger fruit than the strawberries grown at 30/25 °C. *Ledesma, Nakata & Sugiyama (2008)* showed that the air temperature affected the fruit yield and size of 'Nyoho' and 'Toyonoka' which support the findings of *Kadir, Sidhu & Al-Khatib (2006)*. Recent studies reported that the relative humidity is too high (about 90%) for fruit production during strawberry growing season (generally winter), and it could negatively affect the yield (*Ahn et al., 2021*; *Sim et al., 2020*). As such, both air temperature and relative humidity in the greenhouse are important factors in strawberry cultivation, but most studies were performed locally, so generalizability of these study findings may be limited. Thus, comprehensive studies using a large amount of metadata are needed to

accurately quantify the relationship between the management of air temperature and relative humidity and the production of strawberry.

It is very hard for researchers to monitor and record strawberry data from multiple locations across the country. Therefore, the publicly available strawberry data provided in South Korea is a valuable resource for research purposes. Upon our literature review, most studies have used the averages of environmental variables to understand greenhouse conditions. The indoor greenhouse conditions are highly sensitive to outdoor conditions which vary throughout the day. The averages do not take the volatility of time-varying conditions into consideration. Rather than the averages, the percent of time that air temperature and relative humidity are in an optimal range better reflects the greenhouse management. In addition, strawberry yield tends to decrease throughout a harvest season. In this study, the large-scale longitudinal data are analyzed using mixed-effects model to accurately estimate the impact of maintaining air temperature and relative humidity in an optimal range on the trend of yield loss with respect to harvest time (the number of days since harvest began).

## MATERIALS AND METHODS

### Data collection

For a pilot study, an open API dataset was used which was available by a South Korean public database portal (*Ministry of Interior & Safety of South Korea, 2021*). It included 13 farms during the 2017–18 season. For the primary study, a public dataset was used which was available by the South Korean government (*Ministry of Interior & Safety of South Korea, 2022*). It included 78 farms across nine provinces in South Korea during the 2020–21 season. 'Seolhyang', 'Keumsil', and 'Jukhyang' cultivars were included in the data, and 'Seolhyang' was the most commonly cultivated cultivar in the greenhouses. Air and soil temperatures, relative humidity, solar radiation, soil water content, precipitation, $CO_2$ concentration, EC, pH, and the number of fruits produced were collected from each farm. Figure 1 provides the map of South Korea and indicates the strawberry farms included in the pilot and primary studies. Table 1 provides information of the strawberry farms and environmental variables for the primary study, and Table 2 provides the information for the pilot study. Some variables were not recorded for all time periods across the season, but air temperature and relative humidity were recorded for most of the time periods.

For feasible and specific aims of this study, we focused on describing and estimating the relationship between the air temperature and relative humidity and the fruit yield from February to June of 2021. We curated and analyzed data on the air temperature and relative humidity inside greenhouses and the number of fruit yields per plant during the 5-month period. The same variables were available in the pilot data, so we were able to implement the same statistical models to analyze both pilot data (13 farms in the 2017–18 season) and primary data (78 farms in the 2020–21 season).

### Statistical analysis

For statistical modeling, the outcome (dependent) variable of interest was the number of fruits per plant, and the explanatory (independent) variables were air temperature (°C),

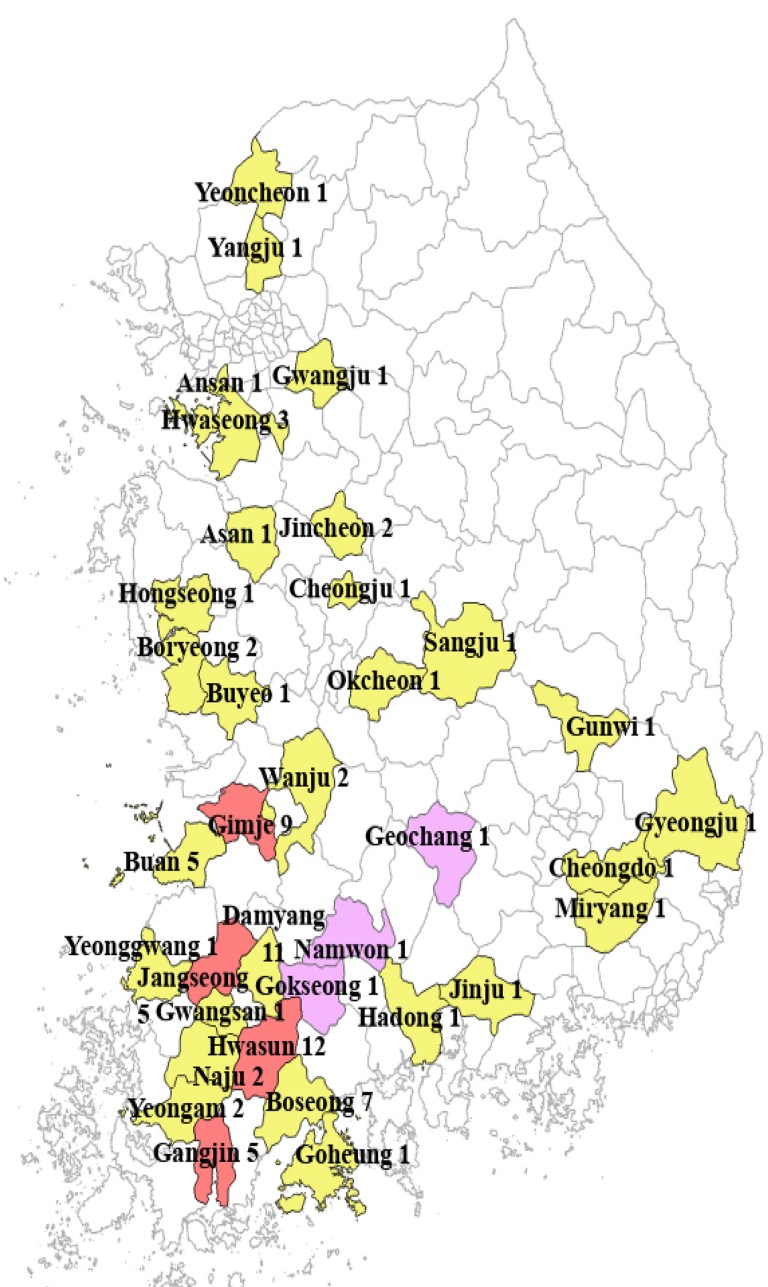

**Figure 1 Distribution of the strawberry farms from which data were collected (yellow: primary dataset; pink: pilot dataset; red: both).**

relative humidity (%), and month. The month is an important variable because the strawberry plant tends to produce less fruit, on average, with respect to harvest time. Hereafter, the month is denoted by M = 2, 3, 4, 5, and 6 for February, March, April, May, and June, respectively. In addition to these observable factors (air temperature, relative humidity, and month), there might be unobserved factors which vary across multiple farms (*e.g.*, greenhouse management skill, equipment). Each farm was repeatedly observed

**Table 1 Information of the strawberry farms from which data were collected.**

| Province | City | No. of farms | Cultivation type | Cultivar | Greenhouse type | Collected environmental data (No. of farms) |
|---|---|---|---|---|---|---|
| Gyeonggi-do | Ansan | 1 | Hydroponic | Seolhyang | Multi-span | EAT, ERH, ESR, EWD, EWS, IAT, IRH, ICC, IST, ISWC |
| | Gwangju | 1 | | | | EAT, ESR, EWS, IAT, IRH, ICC, IST, |
| | Yangju | 1 | | | | EAT, ERH, ESR, IAT, IRH |
| | Yeoncheon | 1 | | | | EAT, ERH, EWD, EWS, IAT, IRH, ICC, IST, ISWC |
| | Hwaseong | 3 | | Seolhyang or Keumsil | Single- or multi-span | EAT (3), ERH (3), EWD (3), EWS (3), IAT (3), IRH (3), IST (2), ISWC (2) |
| Chungcheong nam-do | Buyeo | 1 | Hydroponic | Seolhyang | Single-span | IAT, IRH, ICC |
| | Hongseong | 1 | | | | EAT, ERH, IAT, IRH, IST, ISWC |
| | Asan | 1 | | | Multi-span | EAT, ERH, ESR, EWD, EWS, IAT, IRH, ICC, IST, ISWC |
| | Boryeong | 2 | | | Single- or multi-span | EAT (2), ESR (2), EWS (2), IAT (2), IRH (2), ICC (2) |
| | Cheonan | 1 | Soil | Keumsil | Single-span | EAT, ERH, ESR, EWD, EWS, IAT, IRH, IST, ISWC |
| Chungcheong buk-do | Jincheon | 2 | Hydroponic | Seolhyang | Single-span | EAT (2), ERH (2), ERS (1), EWD (1), EWS (2), IAT (2), IRH (2), ICC (2), ISEC (1), IST (2), ISWC (2) |
| | Okcheon | 1 | | | | IAT, IRH, ICC |
| | Cheongju | 1 | | Keumsil | | EAT, ERH, EWD, EWS, IAT, IRH, ICC |
| Jeolla nam-do | Naju | 2 | Hydroponic | Seolhyang | Single-span | EAT (2), ERH (2), ESR (2), EWD (2), EWS (2), IAT (2), IRH (2) |
| | Yeonggwang | 1 | | | | EAT, ESR, EWS, IAT, IRH, IST |
| | Yeongam | 2 | | | Multi-span | EAT (2), ERH (1), ESR (2), EWD (1), EWS (2), IAT (2), IRH (2), ICC (2), ISEC (1), IST (1), ISWC (1) |
| | Boseong | 7 | | | Single- or multi-span | EAT (5), ERH (5), EWD (5), EWD (5), IAT (7), IRH (7), ICC (6), IST (2), ISWC (2), ISEC (1), ISP (1) |
| | Gangjin | 4 | | | | EAT (2), ESR (2), EWS (2), IAT (4), IRH (4), ICC (2), IST (1), ISWC (2) |
| | Hwasun | 7 | | | | EAT (6), ERH (1), ESR (6), EWD (1), EWS (3), IAT (7), IRH (7), ICC (7), IST (5), ISWC (2) |
| | Jangseong | 4 | | Seolhyang or Keumsil | Single-span | EAT (4), ERH (1), ESR (4), EWD (1), EWS (2), IAT (4), IRH (4), ICC (2), IST (1), ISWC (1) |
| | Goheung | 1 | | Keumsil | Multi-span | EAT, ERH, EWD, EWS, IAT, IRH, ICC, ISR |
| | Damyang | 11 | Hydroponic or Soil | Seolhyang, Keumsil, or Jukhyang | Single-span | EAT (11), ERH (5), ESR (6), EWS (6), IAT (11), IRH (12), ICC (8), IST (3), ISWC (3) |
| Jeolla buk-do | Buan | 5 | Hydroponic | Seolhyang | Multi-span | EAT (5), ESR (5), EWS (5), IAT (5), IRH (5), ICC (5), IST (1), ISWC (1) |
| | Wanju | 2 | | | Single- or multi-span | EAT (1), ESR (1), EWS (1), IAT (2), IRH (2), ICC (1) |
| | Gimje | 6 | | Seolhyang or Keumsil | | EAT (6), ESR (6), EWS (6), IAT (6), IRH (6), ICC (6), IST (2), ISWC (1) |
| Gyeongsang nam-do | Hadong | 1 | Hydroponic | Seolhyang | Single-span | EAT, ESR, EWD, EWS, IAT, IRH, ICC |
| | Jinju | 1 | | | Multi-span | EAT, ESR, EWS, IAT, IRH, ICC, IST |
| | Miryang | 1 | | Keumsil | | EAT, ERH, EWD, EWS, IAT, IRH, ICC, ISR, IST, ISWC |

(Continued)

| Province | City | No. of farms | Cultivation type | Cultivar | Greenhouse type | Collected environmental data (No. of farms) |
|---|---|---|---|---|---|---|
| Gyeongsang buk-do | Cheongdo | 1 | Hydroponic | Seolhyang | Multi-span | EAT, ERH, ESR, EWD, EWS, IAT, IRH, ICC, IST, ISWC |
| | Gyeongju | 1 | | | | EAT, ESR, EWS, IAT, IRH, ICC |
| | Sangju | 1 | | | | EAT, ERH, EWD, EWS, IAT, IRH, ICC, IST, ISWC |
| | Gunwi | 1 | | Unknown | | EAT, ESR, EWD, EWS, IAT, IRH, ICC, IST, ISWC |
| Daejeon-si | Yooseong | 1 | Hydroponic | Seolhyang | Single-span | EAT, ERH, IAT, IRH |
| Gwangju-si | Gwangsan | 1 | Unknown | Seolhyang | Unknown | EAT, ESR, EWD, EWS, IAT, IRH, ICC, IST |

**Note:**

EAT, External air temperature; ERH, External relative humidity; ESR, External solar radiation; EWD, External wind direction; EWS, External wind speed; IAT, Internal air temperature; IRH, Internal relative humidity; ICC, Internal $CO_2$ concentration; ISR, Internal solar radiation; IST, Internal soil temperature; ISWC, Internal soil water content; ISEC, Internal soil electrical conductivity; ISP, Internal soil pH.

**Table 2 Information of the strawberry farms from which data were collected (small data set).**

| Province | City | No. of farms | Cultivar | Collected environmental data (No. of farms) |
|---|---|---|---|---|
| Jeolla nam-do | Gangjin | 1 | Seolhyang | EAT, EWS, IAT, IRH, ICC |
| | Jangseong | 1 | | EAT, EWS, IAT, IRH, ICC |
| | Gokseong | 1 | | EAT, EWS, IAT, IRH, ICC |
| | Hwasun | 5 | | EAT(5), EWS(5), IAT(5), IRH(5), ICC(5) |
| Jeolla buk-do | Namwon | 1 | Seolhyang | EAT, EWS, IAT, IRH, ICC |
| | Gimje | 3 | | EAT(3), EWS(3), IAT(3), IRH(3), ICC(3) |
| Gyeongsang nam-do | Geochang | 1 | Seolhyang | EAT, EWS, IAT, IRH, ICC |

in the dataset, therefore a mixed-effects model was suitable to account for both fixed effects (air temperature, relative humidity, and month) and random effects (farms).

In the datasets, the air temperature and relative humidity were recorded hourly. Using the hourly information, the percent of time that the air temperature was between 15 °C and 20 °C was calculated. Hereafter, this variable is denoted by $T_\%$, and it was used in the mixed-effect model. Similarly, the percent of time that the relative humidity was between 0% and 50%. Hereafter, this variable is denoted by $H_\%$, and it was used in the model. Note that $T_\%$ and $H_\%$ were chosen, instead of the average air temperature and relative humidity, to assess the management of greenhouse environments. High values of $T_\%$ and $H_\%$ imply low volatilities around the optimal ranges which cannot be captured by the averages. For instance, if the air temperature was always 10 °C during the night and was always 30 °C during the day, the daily average would be 20 °C, but $T_\% = 0$.

The number of fruits per plant was transformed using the natural logarithm to respect the normal error assumption, and its average is denoted by μ hereafter. Two mixed-effects models were used to explain μ as a function of $T_\%$, $H_\%$, and M (fixed effects) and farms (random effects). For the first model, denoted by Model 1, the average was specified as $\mu = \beta_0 + \beta_1 M + \beta_2 T_\% + \beta_3 H_\%$. Under this model, the null hypothesis was $H_0: \beta_2 = \beta_3 = 0$, and the alternatively hypothesis was $H_1: \beta_2 > 0$ and $\beta_3 > 0$. In other words, we tested

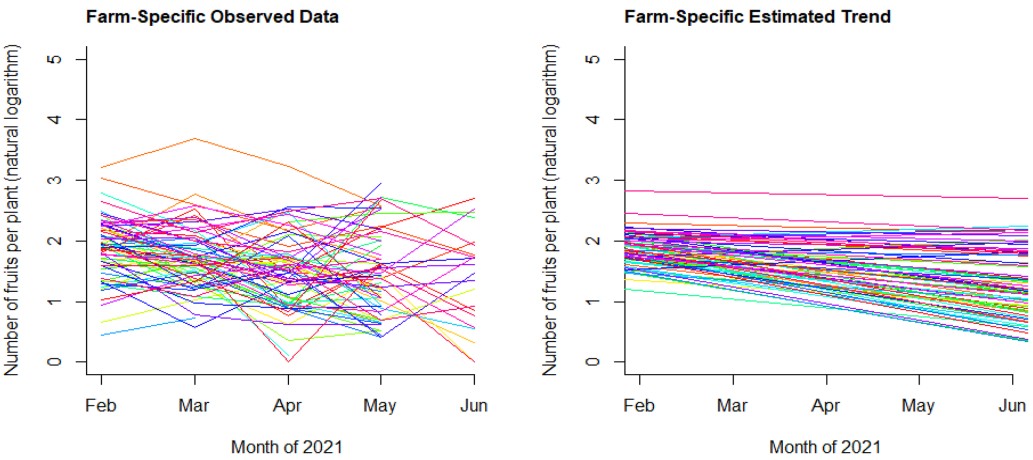

**Figure 2  Observed data (left) and estimated linear trend (right) for each of the 78 farms.**

whether a higher number of fruits is expected when $T_\%$ and $H_\%$ are higher at a given month. For the second model, denoted by Model 2, the average was specified as $\mu = \beta_0 + \beta_1 M + \beta_2 T_\% + \beta_3 H_\% + \beta_4 (M \times T_\%) + \beta_5 (M \times H_\%)$. Under this model, the null hypothesis was $H_0: \beta_4 = \beta_5 = 0$, and the alternative hypothesis was $H_1: \beta_4 > 0$ and $\beta_5 > 0$. In other words, we tested whether the decreasing expected yield over harvest time is mitigated when values of $T_\%$ and $H_\%$ are high.

The number of fruits per plant was also recorded weekly for the most of the evaluation period. For both Model 1 and Model 2, we considered both weekly and monthly average number to observe whether the statistical inference would be sensitive to the choice between the weekly average and the monthly average. For modeling the weekly average, zeroes were removed from the analysis, and M (month) was replaced by W (week). Both Model 1 and Model 2 were fitted to both 2017–18 dataset (13 farms) and 2020–21 dataset (78 farms).

## RESULTS

Figure 2 shows that the expected number of fruits per plant decreased over time (monthly) using the 2020–21 dataset. Each farm has its own management practices and experiences, and the magnitude of negative slopes (*i.e.*, decreasing yield) varied across the farms. Figure 3 shows the management of air temperature and of relative humidity across the farms. On average, the farms maintained the air temperature between 15 °C and 20 °C for about 20% of the time and the relative humidity between 0% and 50% for about 17% of the time during the observation period (February to June). The figure also shows that the management of air temperature and of relative humidity vary across the farms.

Table 3 quantifies the monthly relationship between the expected yield and $T_\%$ and $H_\%$ *via* the regression parameters estimated by the mixed-effects model. The left columns of Table 3 are for the 2020–21 data, and the right columns are for the 2017–18 data. Focusing on the 2020–21 data (78 farms), Model 1 showed that $T_\%$ and $H_\%$ are not related to the monthly expected yield ($p = 0.66$ and 0.94, respectively), but it showed that the monthly

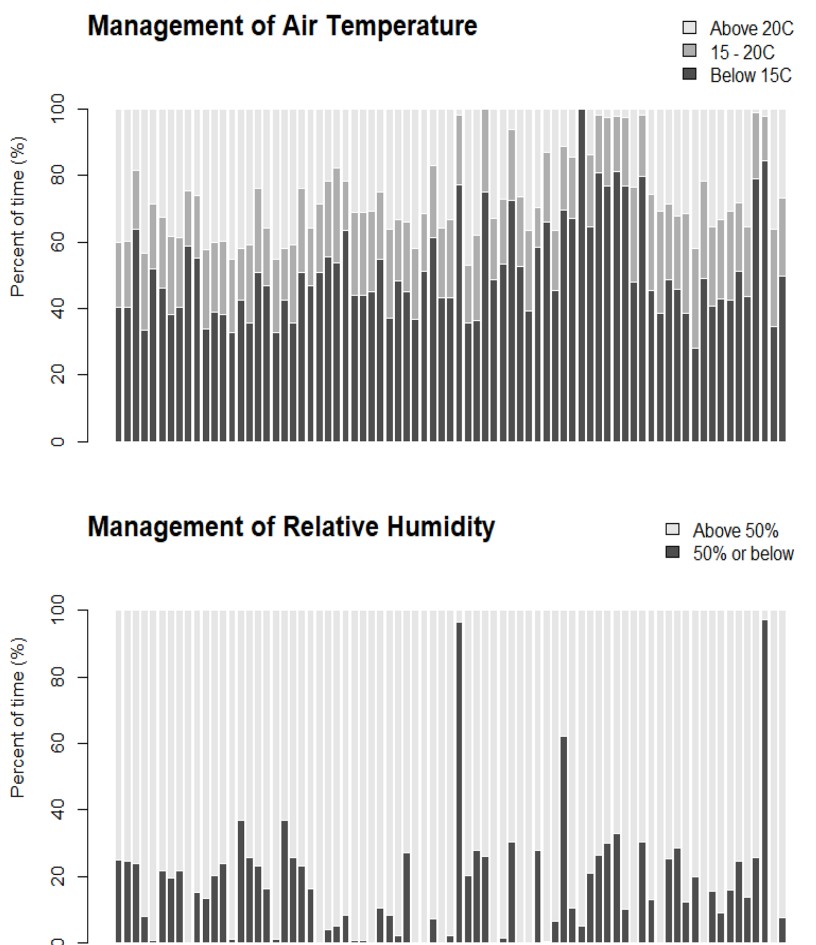

**Figure 3** Percent of time (%) that the temperature was below 15 °C, between 15 °C and 20 °C, and above 20 °C (top panel), and humidity was at 50% or below and above 50% (bottom panel), for each of the 78 farms.

**Table 3 Regression parameters estimated under the mixed-effects model (monthly average).**

| Model | Parameter | 2020–21 season (77 farms) | | | 2017–18 season (13 farms) | | |
|---|---|---|---|---|---|---|---|
| | | Estimate | SE | p-value | Estimate | SE | p-value |
| 1 | $\beta_0$: Intercept | 2.1006 | 0.1002 | <0.0001 | 1.8091 | 0.2271 | <0.0001 |
| | $\beta_1$: M | −0.1479 | 0.0361 | 0.0001 | 0.0298 | 0.0265 | 0.2634 |
| | $\beta_2$: $T_\%$ | 0.0021 | 0.0047 | 0.6552 | −0.0301 | 0.0066 | <0.0001 |
| | $\beta_3$: $H_\%$ | 0.0001 | 0.0021 | 0.9444 | 0.0078 | 0.0166 | 0.6421 |
| 2 | $\beta_0$: Intercept | 2.6958 | 0.2201 | <0.0001 | 3.5517 | 0.3972 | <0.0001 |
| | $\beta_1$: M | −0.3216 | 0.0659 | <0.0001 | −0.2253 | 0.0551 | 0.0001 |
| | $\beta_2$: $T_\%$ | −0.0210 | 0.0119 | 0.0802 | −0.0986 | 0.0162 | <0.0001 |
| | $\beta_3$: $H_\%$ | −0.0121 | 0.0046 | 0.0097 | −0.0538 | 0.0597 | 0.3696 |
| | $\beta_4$: M × $T_\%$ | 0.0059 | 0.0027 | 0.0295 | 0.0097 | 0.0021 | <0.0001 |
| | $\beta_5$: M × $H_\%$ | 0.0037 | 0.0012 | 0.0022 | 0.0092 | 0.0076 | 0.2288 |
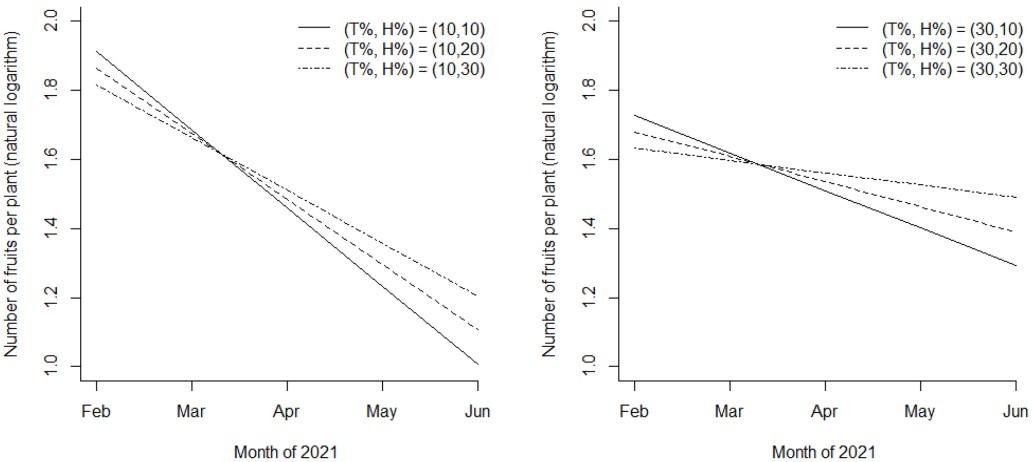

**Figure 4 Estimated average of the logarithmic number of fruit per plant, for the six scenarios: (T%, H%) = (10, 10), (10, 20), and (10, 30) on the left and (T%, H%) = (30, 10), (30, 20), and (30, 30) on the right.**

expected yield decreased with respect to month ($p = 0.0001$). Model 2, however, revealed that both $T_\%$ and $H_\%$ are related to the slope of yield with respect to month ($p = 0.03$ and 0.0022, respectively). Both beta4 and beta5 were estimated positively, and it is interpreted that higher $T_\%$ and $H_\%$ mitigated the magnitude of the negative slope of yield with respect to month. The similar trend was observed in the pilot 2017–18 data (13 farms). The statistical significance for the interaction between $T_\%$ and month was stronger in the 2017–18 data ($p < 0.0001$) than in the 2020–21 data ($p = 0.03$), and the interaction between $H_\%$ and month was weaker in the 2017–18 data ($p = 0.23$) than in the 2020–21 data ($p = 0.0022$).

We note that Model 1 and Model 2 have different objectives. Model 1 explains the expected strawberry yield using $T_\%$, $H_\%$, and M assuming the slope of the yield with respect to M is constant given $T_\%$ and $H_\%$, whereas Model 2 assumes that the slope of the yield with respect to M depends on $T_\%$ and $H_\%$. The results of 2020–21 data indicate that, when compared to Model 1, Model 2 better explains the impact of the management of air temperature and relative humidity on the yield. In other words, while the expected yield decreases as the plant continues to produce the fruit, high values of $T_\%$ and $H_\%$ are important for mitigating the decreasing yield with respect to month.

Figure 4 shows the expected yield (the log-transformed average number of fruits per plant) at $T_\% = 10$ and 30 and $H_\% = 10$, 20, and 30 using the regression parameters estimated by the 2020–21 data. According to the model estimates, the expected yield would reduce substantially over time when $T_\% = 10$ and $H_\% = 10$ (the left panel of Fig. 4), whereas it would be well maintained for the 5-month period when $T_\% = 30$ and $H_\% = 30$ (the right panel of Fig. 4). For instance, over the 5-month period, the median number of fruits per plant reduced by 50% when $T_\% = 10$ and $H_\% = 10$, and the reduction was only 9% when $T_\% = 30$ and $H_\% = 30$.

**Table 4 Regression parameters estimated under the mixed-effects model (weekly average).**

| Model | Parameter | 2020–21 season (77 farms) | | | 2017–18 season (13 farms) | | |
|---|---|---|---|---|---|---|---|
| | | Estimate | SE | p-value | Estimate | SE | p-value |
| 1 | $\beta_0$: Intercept | 2.0592 | 0.0673 | <0.0001 | 2.1712 | 0.0813 | <0.0001 |
| | $\beta_1$: W | −0.0025 | 0.0009 | 0.0043 | −0.0023 | 0.0028 | 0.4221 |
| | $\beta_2$: $T_\%$ | 0.0017 | 0.0021 | 0.4120 | −0.0133 | 0.0027 | <0.0001 |
| | $\beta_3$: $H_\%$ | −0.0010 | 0.0013 | 0.4500 | −0.0063 | 0.0041 | 0.1317 |
| 2 | $\beta_0$: Intercept | 2.3449 | 0.1135 | <0.0001 | 2.5613 | 0.1445 | <0.0001 |
| | $\beta_1$: W | −0.0055 | 0.0013 | <0.0001 | −0.0212 | 0.0063 | 0.0009 |
| | $\beta_2$: $T_\%$ | −0.0098 | 0.0056 | 0.0794 | −0.0328 | 0.0078 | <0.0001 |
| | $\beta_3$: $H_\%$ | −0.0079 | 0.0032 | 0.0137 | −0.0297 | 0.0137 | 0.0307 |
| | $\beta_4$: $M \times T_\%$ | 0.0001 | <0.0001 | 0.0228 | 0.0008 | 0.0003 | 0.0060 |
| | $\beta_5$: $M \times H_\%$ | 0.0001 | <0.0001 | 0.0094 | 0.0012 | 0.0006 | 0.0524 |

Finally, Table 4 summarizes the model estimates for the weekly average number of fruits, and the similar patterns were observed when compared to the monthly average number of fruits for the two seasons (Table 3).

## DISCUSSION

Past studies have shown that environmental variables controlled in greenhouses during strawberry cultivation can affect both growth and fruit yield (*Ahn et al., 2021*; *Sim et al., 2020*). *Sim et al. (2020)* predicted strawberry growth and fruit yield by air and soil temperatures, relative humidity, soil moisture content, EC, photosynthetic active radiation, and vapor pressure deficit. These environmental variables predicted the growth and fruit yield with high correlation coefficients. However, it is not easy to collect all environmental variables from all commercial farms due to high cost and low need. Some farms may be unable to install certain sensors due to structural issues. The challenges of missing variables were observed in the dataset (Tables 1 and 2). Therefore, the most widely used and influential environmental variables were selected for this study. Herein, daily air temperature and relative humidity were considered as the main environmental factors for the following four reasons. First, the effect of air temperature and relative humidity on the number of fruit yield has been extensively evaluated in a variety of fruits and vegetables including strawberry (*Demirsoy et al., 2007*; *Ledesma & Sugiyama, 2005*), tomato (*Abdalla & Verkerk, 1968*; *Harel et al., 2014*), and sweet pepper (*Bakker, 1989*; *Khah & Passam, 1992*). Second, as aforementioned, the two factors are easily observable in all farms, hence the statistical models are applicable to most farms (Tables 1 and 2). Third, daily air temperature and relative humidity reflect other key environmental factors due to their high correlations. *Ahn et al. (2021)* and *Jo et al. (2021)* showed that the trend in air temperature was similar to that of soil temperature and photosynthetic active radiation, and the trend in relative humidity was related to vapor pressure deficit. Lastly, in our previous study, the correlation coefficients between daily air temperature and relative humidity and strawberry fruit yield were 0.82 and −0.93, respectively (*Sim et al., 2020*).

Most studies have considered the averages of air temperature and relative humidity to explain an outcome variable. However, the number of fruit yield was not significantly related to the averages of air temperature and relative humidity in this study. Instead, we used the statistical models after considering the following. First, we chose an optimal air temperature range—between 15 °C and 20 °C—and evaluated the management of air temperature by estimating the percent of time that air temperature was within the range for each farm. The observed air temperature ranged between 5 °C and 40 °C across the farms, and the average of air temperature might not be an accurate measure of the management of air temperature. We considered that good management entails reducing the volatility around an ideal air temperature, if such an ideal point exists. Strawberry growth is optimal at the air temperature of 23–28 °C during daytime and 5–10 °C during nighttime (*Takei, 2010*). On the other hand, a low air temperature changes strawberry fruit size and color and potentially damages strawberry fruit, and a high air temperature potentially reduces photosynthetic rate, fruit yield, and sugar content of the fruit (*Ariza et al., 2012*; *Wang & Camp, 2000*). In this study, the air temperature of 15–20 °C is chosen based on suggestions in literature (*Bish, Cantliffe & Chandler, 2002*; *Kadir, Sidhu & Al-Khatib, 2006*). Second, the winter in South Korea is dry, but the relative humidity in greenhouses is very high (90% or above). Hence, farmers should reduce the internal relative humidity to facilitate transpiration and prevent diseases. As shown in Fig. 2, many farms observed in this study struggled with maintaining a low relative humidity inside greenhouses. Even though a suggested range of relative humidity is 60–80% (*Choi, Chung & Suh, 1997*), we evaluated the management of relative humidity at 50% or below. This is because relative humidity decreases rapidly when ventilation is applied and it rises immediately after the ventilation. In this regard, the percent of time that relative humidity is 50% or below ($H_\%$) reflects the level of effort to lower the relative humidity during a high-humidity season in South Korea. Therefore, the results should be generalized with care. It does not necessarily imply that the driest conditions are optimal, as fogging could be beneficial in a dry environment (*Morgan, 2006*). Third, though all farms have different expected fruit yield for various reasons, the gradually decreasing trend of fruit yield throughout a harvest season is a natural phenomenon observed in most farms (Fig. 2). In this regard, it was reasonable to hypothesize that the downtrend can be mitigated by managing the indoor air temperature and relative humidity, and we used the statistical models to explain the "slope" of fruit yield with respect to harvest time (the number of days since harvest began). Furthermore, the mixed-effects model accounts unobserved variables such as leaf mass per unit leaf area and light intensity (*Bertin & Gary, 1998*; *Chatterton, Lee & Hungerford, 1972*; *Reddy et al., 1989*), $CO_2$ concentration, and air temperature (*Bertin & Gary, 1998*; *Acock, Charles-Edwards & Sawyer, 1979*; *Charles-Edwards, 1979*; *Leadley & Reynolds, 1989*). Resultantly, it is more reasonable to conclude that the impact of air temperature and relative humidity is especially significant at the later phase of a harvest season rather than concluding that the impact is constant throughout the harvest season (Tables 3 and 4).

## CONCLUSIONS

Based on publicly available data collected from a large number of farms across South Korea, we evaluated the management of air temperature (15–20 °C) and relative humidity (50% or below) at each greenhouse. Using the percent of time within the optimal ranges, we conclude that the impact of indoor air temperature and relative humidity is especially significant at the later phase of a harvest season. Therefore, strawberry farmers are needed to continually manage air temperature and relative humidity through ventilation to reduce the loss of yield at the end of a harvest season. If a harvest season is in winter like in South Korea, active management such as heating and forced ventilation may be required.

## ACKNOWLEDGEMENTS

We would like to thank Editage for English language editing.

### Funding

This research was funded by the National Research Foundation of Korea, Grant number 2019R1I1A3A01063693. The funders had no role in study design, data collection and analysis, decision to publish, or preparation of the manuscript.

### Grant Disclosures

The following grant information was disclosed by the authors:
National Research Foundation of Korea: 2019R1I1A3A01063693.

### Competing Interests

The authors declare that they have no competing interests.

### Author Contributions

- Steven Kim conceived and designed the experiments, performed the experiments, analyzed the data, prepared figures and/or tables, authored or reviewed drafts of the article, and approved the final draft.
- Jung Su Jo performed the experiments, analyzed the data, prepared figures and/or tables, and approved the final draft.
- Vicky Luk performed the experiments, analyzed the data, prepared figures and/or tables, and approved the final draft.
- Sung Kyeom Kim conceived and designed the experiments, authored or reviewed drafts of the article, and approved the final draft.
- Dong Sub Kim conceived and designed the experiments, prepared figures and/or tables, authored or reviewed drafts of the article, and approved the final draft.

## Data Availability

The data used in this article is publicly available at:

- Ministry of Interior And Safety. 2021. Public data portal. Rural Development Administration_Smart farm excellent farmhouse open data. https://www.data.go.kr/data/15042594/openapi.do.

- Ministry of Interior And Safety. 2022. Public data portal. Agriculture, Forestry and Fisheries Food Education and Culture Information Service_Smart Farm Big Data. https://www.data.go.kr/data/15015449/openapi.do.

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
