# Peer review of "Estimating the impact of environmental management on strawberry yield using publicly available agricultural data in South Korea"

_PeerJ, doi:10.7717/peerj.15390_

## Round 0.1 · original submission · Major Revisions

Dear authors, please consider all reviewers' comments and explain the reasons should you decide not to make some of the suggested changes to the manuscript. The English language should be improved.

Reviewer 1 ·

Basic reporting

no comment

Experimental design

no comment

Validity of the findings

I didn't think the findings of this paper were very novel. For many journals, the low impact/value of the findings would not merit publication. Simply saying that maintaining temperature and RH within optimum ranges is expected.

Additional comments

All my editorial comments were made on the PDF.

Annotated reviews are not available for download in order to protect the identity of reviewers who chose to remain anonymous.

Reviewer 2 ·

Basic reporting

The English language can be improved to ensure that readers can clearly understand your text. Some examples where the language could be improved
include lines 194,202,203 etc
A more detailed literature review can be provided such as that helped to select the optimum temperature range

Experimental design

Primary research is on finding out the importance of air temperature and relative humidity maintenance in strawberry yield. the theme of the research is relevant. However, the novelty and relevance of this study should be elaborated explicitly in the introduction.


The statistical method is seldom described in the paper. A holistic description of the methods used in the study will be beneficial to the readers.

The selection criteria of the optimum temperature range are not given in the text apart from a statement in line 225. It needs more explanation as it is critical to the paper.

Validity of the findings

As per table 3, the SE is comparatively higher than the estimate (especially in case of 2020-21).Then, how can these estimates be statistically relevant considering that for 2020-2021 season the number of farms is also not less as compared to 2017-2018.

---

## Round 0.2 · accepted · Accept

The authors considered all the comments and suggestion made by the reviewers. I see the manuscript improved and ready to be accepted.